# Lipid Composition Analysis and Characterization of Acyl Sterol Glycosides in Adzuki and Soybean Cultivars by Non-Targeted LC-MS

**DOI:** 10.3390/foods12142784

**Published:** 2023-07-21

**Authors:** Rachana M. Gangadhara, Siddabasave Gowda B. Gowda, Divyavani Gowda, Ken Inui, Shu-Ping Hui

**Affiliations:** 1Graduate School of Global Food Resources, Hokkaido University, Kita-9, Nishi-9, Kita-Ku, Sapporo 060-0809, Japan; rachanachem03@gmail.com; 2Faculty of Health Sciences, Hokkaido University, Kita-12, Nishi-5, Kita-ku, Sapporo 060-0812, Japan; divyavani@hs.hokudai.ac.jp; 3HIRYU Co., Ltd., Chuo-Cho 2-32, Kashiwa-shi 277-0021, Japan; kemist20@gmail.com

**Keywords:** liquid chromatography, mass spectrometry, bean samples, lipid analysis, cluster correlation

## Abstract

Beans, a globally significant economic and nutritional food crop, are rich in polyphenolic chemicals with potential health advantages, providing high protein, fiber, minerals, and vitamins. However, studies on the global profiling of lipids in beans are limited. We applied a non-targeted lipidomic approach based on high-performance liquid chromatography coupled with linear ion trap–Orbitrap mass spectrometry (HPLC/LTQ-Orbitrap-MS) to comprehensively profile and compare the lipids in six distinct bean cultivars, namely, adzuki red beans—adzuki cultivar (ARB-AC), adzuki red beans—Benidainagon cultivar (ARB-BC), adzuki red beans—Erimoshouzu cultivar (ARB-EC), soybean—Fukuyutaka cultivar 2021 (SB-FC21), soybean—Fukuyutaka cultivar 2022 (SB-FC22), and soybean—Oosuzu cultivar (SB-OC). MS/MS analysis defined 144 molecular species from four main lipid groups. Multivariate principal component analysis indicated unique lipid compositions in the cultivars except for ARB-BC and ARB-EC. Evaluation of the concentrations of polyunsaturated fatty acid to saturated fatty acid ratio among all the cultivars showed that SB-FC21 and SB-FC22 had the highest value, suggesting they are the most beneficial for health. Furthermore, lipids such as acyl sterol glycosides were detected and characterized for the first time in these bean cultivars. Hierarchical cluster correlations revealed the predominance of ceramides in ARB-EC, lysophospholipids in SB-FC21, and glycerophospholipids in SB-OC. This study comprehensively investigated lipids and their compositions in beans, indicating their potential utility in the nutritional evaluation of beans as functional foods.

## 1. Introduction

Beans (*Phaseolus vulgaris*) are members of the Leguminosae, which are essential sources of protein, energy (starch), and dietary fiber for millions of people [1]. About 20 kinds of leguminous plants are used in significant quantities as dry grains for human nourishment [2]. Complex carbohydrates, vitamins, minerals, protein, fiber, resistant starch, and phytochemicals with multiple bioactive properties are all found in beans with low glycemic index [1,3]. Adzuki bean (*Vigna angularis*) is a smaller, reddish-brown bean popular in Japan and other Asian countries. It is a popular food for humans because it is high in carbohydrates, digestible protein, minerals, and vitamins [4,5,6]. Adzuki beans include a variety of flavonoids and phenolic substances with antioxidant, antibacterial, and anti-inflammatory activities [7,8,9]. These compounds can aid in reducing oxidative stress and inflammation in the body, hence contributing to the prevention of hypolipidemic and chronic illnesses such as heart disease, diabetes, and cancer [8,10].

Soybean (*Glycine max* (L.)) is generally considered an oil seed and is gaining attention because it is rich in various natural antioxidants, including β-carotene and tocopherols, and has high protein content [11,12]. China and Japan are the largest importing countries of beans and its products [13]. Soybeans are high in bioactive components, including isoflavones, which are phytoestrogens that can help lower the risk of breast cancer and other hormone-related malignancies [14,15]. Soybeans also include saponins, compounds that have been demonstrated to decrease cholesterol. By reducing low-density lipoprotein (LDL) levels in the blood, these compounds can help to minimize the risk of heart disease [16]. Additionally, soybeans may help to improve bone density, which can prevent osteoporosis [17]. Human studies have implied the beneficial effects of soy proteins on serum lipids and colorectal cancer in postmenopausal women [18]. Studies discovered that consuming adzuki beans increased insulin sensitivity and glucose metabolism in obese rats [10], decreased blood pressure, and improved lipid profiles in hypertensive rats. These findings also imply that consuming adzuki beans may help lower the risk of type 2 diabetes and cardiovascular disease [4,19]. Furthermore, the presence of phytochemicals and their contents in the red-coated adzuki bean provides excellent health benefits [9]. Untargeted metabolomics research on fermented adzuki beans using liquid chromatography–mass spectrometry (LC-MS) revealed many alkaloids, organic acids, phenolic compounds, and amino acids [20].

Past analytical studies on lipid composition in beans are limited to specific classes. In a previous study by Yoshida et al., analytical methods such as thin-layer chromatography (TLC) and gas chromatography were utilized to profile lipids in adzuki beans [6]. However, this study was also limited to specific classes of lipids, such as fatty acids, triacylglycerols, and glycerophospholipids. They found that phospholipids and triacylglycerols were predominant components in adzuki beans, accounting for 74% and 13% of total lipids, respectively. Despite the authors making an effort to characterize the adzuki bean lipidome, the investigation of lipids was limited to the molecular species level and the TLC technique employed is less sensitive and requires a large sample. Similarly, advanced targeted LC-MS was utilized by Lee et al. to determine phospholipid contents in 12 soybean cultivars [21]. However, the authors selected only five phospholipid species and quantified them in soybean cultivars of Korean origin by multiple reaction monitoring.

To our knowledge, no studies have reported on comprehensive lipid profiling of beans using the LC-MS technique. Hence, we used extremely sensitive high-performance liquid chromatography combined with linear ion trap–Orbitrap mass spectrometry (HPLC/LTQ-Orbitrap-MS) for lipid bean profiling. By utilizing this advanced technique, we were able to study and compare the correlations of multiple lipids among six distinct cultivars of beans. The nutritional significance of these lipids and acyl sterol glycosides (ASG) lipids were characterized for the first time by analyses of accurate masses and MS/MS fragments.

## 2. Materials and Methods

### 2.1. Sample Collection

Three different cultivars of adzuki bean (*Vigna angularis*), namely, Benidainagon (ARB-BC) cultivated in 2022 from Aomori, Erimoshouzu (ARB-EC) cultivated in 2021 from Hokkaido, adzuki (ARB-AC) cultivated in 2022 from Kagawa, and three different cultivars of soybean (*Glycine max* (L.)), namely, Fukuyutaka cultivated in 2021 (SB-FC21) from Kagawa, Fukuyutaka cultivated in 2022 (SB-FC22) from Kagawa, and Oosuzu cultivated in 2022 (SB-OC) from Aomori, were supplied by HIRYU Co., Ltd. (Kashiwa-Shi, Japan) and cultivated under chemical-free conditions. Beans were crushed in a mixer, and the dry powder was used for total lipid extraction.

### 2.2. Materials

LC-MS grade methanol, isopropanol, and chloroform were purchased from Wako Pure Chemical Industries, Ltd. (Osaka, Japan). Ammonium acetate (1 M solution) was obtained from Sigma-Aldrich (St. Louis, MO, USA). Oleic acid-d9 and EquiSPLASH Lipidomix, quantitative standards for mass spectrometry, were purchased from Avanti Polar Lipids (Ablabaster, AL, USA). 100 μL of premixed EquiSPLASH lipidomix (1 µg/mL) and oleic acid-d9 (10 µg/mL) in methanol were used as internal standard during lipid extraction.

### 2.3. Lipid Extraction

Four to five ceramic beads were added to a 2 mL Eppendorf tube preloaded with 150 mg of crushed bean powder. The samples were homogenized for 1 min (30 s × 2) with the Fisher Scientific Bead mill 4 (Fisherbrand, Tokyo, Japan) homogenizer after adding 1.5 mL of methanol. A modified Bligh–Dyer technique was utilized for total lipid extraction [22,23]. The 100 µL of internal standard mixture (EquiSPLASH lipidomix (1 µg/mL) and oleic acid-d9 (10 µg/mL) in methanol) was added for every 200 µL of homogenized extracts taken in a 1.5 mL Eppendorf tube. Following that, 150 µL of chloroform and 30 µL of Milli-Q (ultrapure) were added to the mixture, vortexed at 3500 rpm for 10 min, and centrifuged at 15,000 rpm at 4 °C for 5 min. The single-phase centrifugate was transferred to a new vial and evaporated under vacuum at 4 °C using a centrifuge evaporator. After complete evaporation, samples were redissolved in 100 µL of methanol, vortexed at 3500 rpm for 1 min, centrifuged at 4 °C at 15,000 rpm for 10 min, and the centrifugate transferred to LC-MS vials for further analysis.

### 2.4. Analysis by LC-MS

The LC-MS conditions used in this experiment were same as in our previous studies [24]. In brief, the lipid composition analysis of bean samples was carried out using the Prominence HPLC system (Shimadzu Corporation, Kyoto, Japan) with a standard autosampler and an oven. Separation was performed using an Atlantis T3 C18 column (2.1 × 150 mm, 3 mm, Waters, Millford, CT, USA) at a 200 µL/min flow rate with a 40 °C column temperature. The volume of sample injected for each run was set to 10 µL. The HPLC consists of a three-pump system and had mobile phases of solvent A with 10 mM aqueous CH_3_COONH_4_, solvent B isopropanol, and solvent C methanol. A linear gradient elution was applied as follows: initially with 30% B and 35% C, with the percentage of solvent B linearly increased to 80% and solvent C decreased to 10% in 14 min, and then the percentage of B increased to 85% in 14 min with constant solvent C. The column was equilibrated for additional 3 min with initial conditions.

Mass spectrometry analysis was performed using an LTQ Orbitrap mass spectrometer (Thermo-Fisher Scientific Inc., San Jose, CA, USA) with data-dependent acquisition conditions (in negative ionization mode) and the parameters were identical to those of our previous reports [25]. The MS conditions were as follows. Negative electrospray ion source voltage was 3.0 kV. Full-scan MS was set at *m*/*z* 160–1900 in Fourier-transform mode with a mass resolution of 60,000. Low-resolution MS^2^ and MS^3^ spectra were acquired by collision-induced dissociation at a collision energy of 40 and 45 V, respectively, with an isolation width of *m*/*z* 3. MS-DIAL software version 4.9 was used to process the raw data for alignment, peak extraction, identification, and peak area integration [24]. MS and MS/MS spectra were used to verify the identification of lipid molecular species. The peak area ratios of the annotated lipids to the internal standard were multiplied by the concentration of added internal standard to achieve relative quantification of lipid molecular species. The concentrations were normalized by the weight of bean samples used for lipid extraction.

### 2.5. Statistical Analysis

The data were represented as the mean and standard deviation (*n* = 5) in Microsoft Excel 2019 and GraphPad Prism 8 software (San Diego, CA, USA). MetaboAnalyst version 5.0 (https://www.metaboanalyst.ca/) was used to perform sparse partial least squares discriminant and cluster correlation analysis (accessed on 9 July 2022). A value of *p* < 0.05 was considered statistically significant (one-way ANOVA with Tukey’s multiple comparisons test).

## 3. Results and Discussion

### 3.1. Distribution of Lipid Classes in Six Different Cultivars of Beans

Bean lipidomic profiles were assessed using HPLC/LTQ-Orbitrap-MS in negative ionization mode. The MS-DIAL application was used to annotate 144 lipids based on their exact masses and MS/MS spectral data. Figure 1A provides a visual representation of the bean cultivars selected for this study. Following identification and relative quantification of lipids by the internal standard method, multivariate analysis was carried out, including one-way ANOVA (*p* < 0.05) with Kruskal–Wallis analysis. Figure 1B shows that all detected lipids in negative ionization mode were statistically significant (*p* < 0.05). The sparse partial least squares discriminant analysis (sPLSDA) revealed differences between the bean cultivars based on changes in their lipidome, as shown in Figure 1C. The variable importance in projection (VIP) identified essential lipids for distinguishing between the cultivars. Component 1 and Component 2 accounted for 82.3% of the total variance in the model, with Component 1 explaining a majority of the variance (54.3%). The lipid composition of all six cultivars of beans exhibited distinct group-specific clustering. Among the cultivars, ARB-BC and ARB-EC exhibited similar lipid compositions, while ARB-AC, SB-FC21, SB-FC22, and SB-OC had distinct compositions compared to these two groups. Positive or negative loading scores that are higher suggest that a variable has a significant impact on the components. Lipid molecular species such as stigmasterol hexosides SG 28:1; O; Hex, SG 29:1; O; Hex, and phosphatidylglycerol: PG (18:2/18:2) primarily contributed to the group separation and had significant positive loading scores.

### 3.2. Concentrations, P:S Ratio, and Hierarchical Cluster Analysis of Free Fatty Acids

An untargeted analysis of bean samples revealed the detection of sixteen free fatty acid (FA) molecular species. The concentration of identified FAs in bean cultivars is shown in Figure 2A. The highest amount of FAs was observed in SB-FC21, followed by SB-FC22. Figure 2B represents the ratio of polyunsaturated fatty acids (PUFAs) to saturated fatty acids (SFAs). A higher P:S ratio is considered to be more beneficial for cardiovascular health, as discussed in our previous studies [24]. The findings indicate that SB-FC21 has the highest P:S ratio, indicating the lowest cardiovascular disease risk among all the cultivars, whereas ARB-BC has the lowest P:S ratio. The hierarchical cluster analysis (HCA) of FAs is depicted in Figure 2C based on Ward clustering and Euclidean distance. It reveals a clear clustering pattern among the different cultivars of beans. The analysis indicates that certain FAs exhibit positive associations with high levels, as represented by red. Conversely, other FAs display negative associations with low levels, as indicated by blue. The fatty acid profiles of the three cultivars of adzuki beans differ from those observed in previous studies on various adzuki bean cultivars [6,26,27]. This could be due to various factors, including cultivar varieties and the method of analysis performed. Moreover, previous studies aimed at determining the fatty acid compositions in complex phospholipids, rather than free fatty acid levels.

### 3.3. Hierarchical Cluster Correlation Analysis of Sphingolipids and Lysophospholipids Characterized in Bean Cultivars

The concentration of ceramide, lysophosphatidic acid (LPA), and lysophosphatidylethanolamine (LPE) identified and their HCA results are shown in Figure 3A,B, respectively. The results show that ARB-AC, ARB-BC, and ARB-EC cultivars contain abundant ceramides, whereas LPA and LPE are higher in SB-FC21 and SB-FC22 cultivars. Ceramides play a crucial role as integral components of eukaryotic cell membranes, contributing to their structural stability. In addition to their structural function, ceramides also function as bioactive lipids involved in numerous cell signaling pathways. These pathways include apoptosis, inflammation, cell cycle arrest, and the heat shock response [28,29,30]. Ceramides serve as a new biomarker in various disorders, including cancer, multiple sclerosis, type 2 diabetes (T2D), Alzheimer’s disease, and coronary artery disease (CAD) [31]. Plant-based ceramides are being widely used in the cosmetics industry. Our analysis of the results showing an abundance of ceramides in adzuki beans suggests that the beans could be a potential source for natural ceramides. The best-studied signaling lysophospholipid is LPA [32]. Lysophosphatidylcholine (LPC), LPC 18:0, LPC 18:2, are abundant in soybeans (SB-FC21/SB-FC22) and have been shown to be a key source of neural membrane biogenesis [33]. The presence of ceramide lipids in different types of beans, including soybean, and their characterization has already been reported [34,35,36]. However, the composition of lipids varies among the different cultivars.

### 3.4. Hierarchical Cluster Correlation Analysis of Glycerophospholipids, Glycerolipids and Sterols Characterized in Bean Cultivars

The HCA of glycerophospholipids (GPs) characterized in bean cultivars such as phosphatidylcholine (PC), phosphatidylglycerol (PG), phosphatidylinositol (PI), and phosphatidylserine (PS) is shown in Figure 4 and Figure 5. The PC, PG, PI, and PS were higher in soybean cultivars SB-OC, SB-FC22, and SB-FC21 than adzuki bean cultivars AB-AC, AB-BC, and AB-EC. PEs are involved in membrane fusion, and the breaking of contractile rings during cell division is an essential role of these lipids in forming cell membranes [37]. PIs are vital for lipid signaling, cell signaling, and membrane trafficking [38], whereas PGs are glycerol-based phospholipids that are significant components of biological membranes [39]. PAs, also known as anionic phospholipids, play a crucial role in cell signaling and metabolic management. PCs are also pulmonary surfactants and essential components of cellular membranes [40]. PCs were the most abundant GPs characterized in adzuki bean samples by Yohida H. et al. [6]. Our analysis results showed an abundance of PCs in soybean cultivars over adzuki beans.

Therefore, considering the abundance of GPs in SB-OC, SB-FC22, and SB-FC21, these three cultivars may provide dietary advantages. Interestingly ARB-BC cultivar showed higher abundance of PEs with linoleic acid (FA 18:2) and linolenic acid (FA18:3). The glycerolipids, sulfoquinovosyl diacylglycerol (SQDG) and sterols, sterol glycosides (SGs) are relatively higher in soybean cultivars (SB-FC 21/22) than adzuki cultivars, as shown in Figure 5. In this study, we identified a greater variety of phospholipid subclasses in different cultivars of beans than previous findings from bean phospholipid analysis [6,21,26,41].

### 3.5. Characterization of Acyl Sterol Glycosides (ASGs) in Bean Cultivars

LC-MS assessment of lipid extracts from diverse bean cultivars confirmed the presence of acyl sterol glycosides (ASGs). These ASG lipids belong to the sterol lipid family and the sugar moiety is acylated with a fatty acid in the C-6 hydroxyl group, thus forming an ASG. One study suggested that ASGs can potentially reduce cholesterol absorption in mice [42]. It was observed that while the acyl group was cleaved from the ASG in the digestive tract, the glycosidic bond remained intact. These findings indicate that both ASG and SG could be beneficial dietary components for lowering cholesterol and reducing the risk of cardiovascular diseases [43,44]. These lipids are predominantly found in plants, including legumes. While ASGs were previously identified in beans [43], their molecular species levels have not been confirmed. Previous studies by Rozenberg et al. and Liu et al. successfully characterized ASG lipids in spelt, wheat and green coffee, respectively [45,46]. However, there have been no reports on the characterization of these lipids, specifically for the bean cultivars investigated in this study.

We detected eight molecular species of ASG in bean samples. We obtained high-resolution masses from MS^1^ spectra and observed fragmentation behavior in low-resolution MS/MS spectra, confirming the presence of these lipids. The list of identified ASG lipids in bean cultivars is provided in Table 1. Figure 6A depicts the extracted ion chromatograms, MS, and MS/MS spectra of all eight ASG molecular species characterized in bean samples. The relative amounts of ASG in each bean cultivar are represented by the heatmap shown in Figure 6B. The ionization of these lipids occurred in negative mode, resulting in the generation of [M+CH_3_COO]^−^ ions as precursor ions. The compound, which elutes with a retention time of 17.22 min and an *m*/*z* value of 859.6682 (C_50_H_88_O_7_ +CH_3_COO^−^, calculated *m*/*z*: 859.6693, error: −1.27 ppm), is fragmented further to produce *m*/*z* 799.4 as [M-H]^−^ ion. Notably, peaks at *m*/*z* 255.2 (FA 16:0) in the MS/MS spectra suggest that the identified compound is ASG 28:1;O;Hex;FA 16:0.

Similarly, the lipid eluted at 16.87 min as *m*/*z* 883.6692 [C_52_H_88_O_7_ +CH_3_COO^−^, calculated *m*/*z* 883.6680, error: 1.35 ppm] and generates *m*/*z* 823.6 as [M-H]^−^ ion. The prominent peaks observed at *m*/*z* 279.3 (FA 18:2) in the MS/MS spectra indicate that the compound is ASG 28:1;O;Hex;FA 18:2. Furthermore, the lipid with a retention time of 17.39 min showcases *m*/*z* 873.6837 [C_52_H_86_O_7_ +CH_3_COO^−^, calculated *m*/*z*: 873.6825, error: 1.37 ppm]. Upon ionization, it gives a peak at *m*/*z* 813.7 as [M-H]^−^ ion.

The presence of peaks at *m*/*z* 255.3 (FA 16:0) in the MS/MS spectra suggests the compound’s identification as ASG 29:1;O;Hex;FA16:0. The compound eluted at 17.22 min as *m*/*z* 871.6673 [C_51_H_88_O_7_ +CH_3_COO^−^, calculated *m*/*z*: 871.6667, error: 0.68 ppm], it ionizes to generate *m*/*z* 811.1 as [M-H]^−^ ion and 255.2 (FA 16:0) in the MS/MS spectra, indicating its identification as ASG 29:2;O;Hex;FA 16:0. Similarly, the lipid eluted at 17.72 min as *m*/*z* 899.6985 [C_53_H_92_O_7_ +CH_3_COO^−^, calculated *m*/*z* 899.6983, error: 0.22 ppm] and ionized to produce *m*/*z* 838.9 as [M-H]^−^ ion and a peak at *m*/*z* 283.3 (FA 18:0) in the MS/MS spectra suggests that the compound is ASG 29:2;O;Hex;FA 18:0.

The compound elutes with a retention time of 16.91 min as an *m*/*z* value of 895.6668 [C_53_H_88_O_7_ +CH_3_COO^−^, calculated *m*/*z*: 895.6666, error: 0.22 ppm], which ionizes further to produce *m*/*z* 834.9 as [M-H]^−^ ion. Notably, the presence of peaks at *m*/*z* 279.3 (FA18:2) in the MS/MS spectra suggests that the identified compound is ASG 29:2;O;Hex;FA 18:2. The lipid eluted at 16.47 min as *m*/*z* 893.6509 [C_53_H_86_O_7_ +CH_3_COO^−^, calculated *m*/*z*: 893.65118, error: −0.31 ppm] and further ionizes to produce *m*/*z* 833.4 as [M-H]^−^ ion; the peaks at *m*/*z* 277.3 (FA 18:3) in the MS/MS spectra suggest that the detected compound is ASG 29:2;O;Hex;FA 18:3. Finally, the lipid eluted at 16.40 min has *m*/*z* 881.6522 [C_52_H_86_O_7_ +CH_3_COO^−^, calculated *m*/*z*: 881.6511, error: 1.24 ppm], and it ionizes to give *m*/*z* 821.8 and *m*/*z* 820.6 as [M-H]^−^ ions respectively. Further, peaks of FA 18:3 (*m*/*z* 277.2) and FA 18:2 (*m*/*z* 279.3) in the MS/MS spectra indicate that it is an isomeric mixture of ASG 28:1;O;Hex;FA 18:3 and ASG 28:2;O;Hex;FA 18:2. Overall, all the MS, MS/MS spectral behaviors match with the MS-DIAL database of ASG lipids.

The lipid molecular species ASG 28:1;O;Hex;FA 18:2 is abundant in SB-FC22, whereas ASG 28:1;O;Hex;FA 16:0, ASG 28:1;O;Hex;FA 18:3 or ASG 28:2;O;Hex;FA 18:2 and ASG 29:1;O;Hex;FA 16:0 were abundant in SB-OC cultivars. Further, ASG 29:2;O;Hex;FA 16:0 and ASG 29:2;O;Hex;FA 18:2 were relatively higher in ARB-BC, whereas ASG 29:2;O;Hex;FA 18:0 and ASG 29:2;O;Hex;FA 18:3 were abundant in ARB-EC cultivars, respectively. Despite the abundances of these ASG lipids being different among different cultivars, to our knowledge, this is the first report on the characterization of these lipid molecular species in adzuki and soybean cultivars. In a previous study, Liu et al., characterized 12 ASG lipid molecular species in green coffee using ultrahigh-performance liquid chromatography–time-of-flight tandem MS in both positive and negative ionization modes. ASG lipids have a strong tendency to ionize both positive and negative modes to give specific diagnostic fragment ions to identify sterol and fatty acid moieties [46]. The mass spectral behavior of ASG lipids characterized in this study was similar to that reported by Li et al. The study has some limitations, including the quantification method used being relative and not absolute. Further the environmental and cultivations conditions may influence the lipid compositions of beans investigated, which were not considered in this study.

## 4. Conclusions

In summary, an HPLC/LTQ–Orbitrap MS method was applied to analyze the lipid compositions and detect previously uncharacterized lipids in different bean cultivars. The P:S ratio was considerably greater in the soybean—Fukuyutaka cultivar 2021 than the adzuki bean cultivars, suggesting the possible health benefit of soybeans over adzuki beans. The cluster correlation analysis revealed a comprehensive profile of different lipid classes and their concentration and distribution in each bean cultivar. Ceramides were discovered to be abundant in adzuki red beans—adzuki cultivar, adzuki red beans—Benidainagon cultivar, and adzuki red beans—Erimoshouzu cultivar, whereas lyosphospholipids were abundant in soybean—Fukuyutaka cultivar 2021. Glycerophospholipids were shown to be more abundant in soybeans than adzuki bean cultivars. For the first time, ASG lipids were identified and characterized in soybean and adzuki bean cultivars by MS/MS analysis. These findings show the distinct and comprehensive lipid compositions of different types of bean cultivars grown naturally in the fields of Japan. Furthermore, lipids in beans can account for their nutritional and health-promoting properties. We intend to apply the untargeted LC/MS technique to explore the lipid compositions in other functional foods.

## Figures and Tables

**Figure 1 foods-12-02784-f001:**
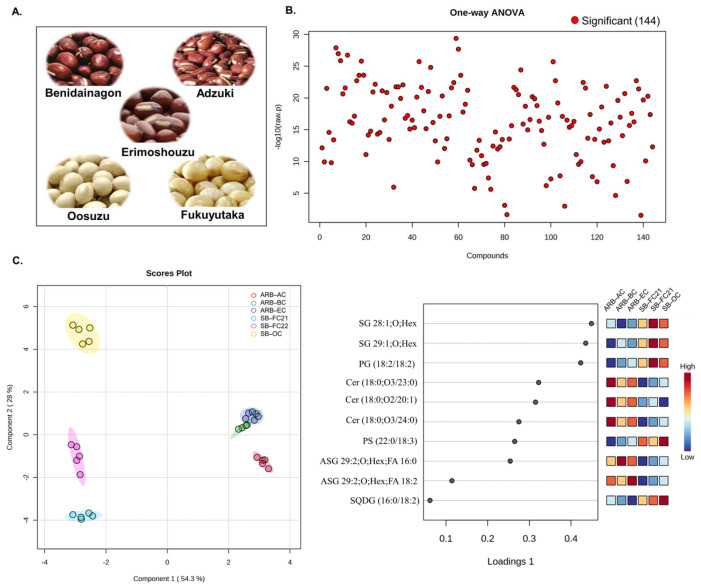
Multivariate analysis of six different cultivars of beans. (**A**) Pictorial representation of bean cultivars. (**B**) One-way ANOVA analysis of 144 identified lipid molecular species (Kruskal–Wallis test with *p* < 0.05). (**C**) Sparse partial least squares discriminant analysis (sPLSDA) score plot and variable importance in projection (VIP) of lipid metabolites. ARB−AC (adzuki red beans—adzuki cultivar), ARB-BC (adzuki red beans—Benidainagon cultivar), ARB−EC (adzuki red beans—Erimoshouzu cultivar), SB−FC21 (soybean—Fukuyutaka cultivar 2021), SB−FC22 (soybean—Fukuyutaka cultivar 2022), SB−OC (soybean—Oosuzu cultivar).

**Figure 2 foods-12-02784-f002:**
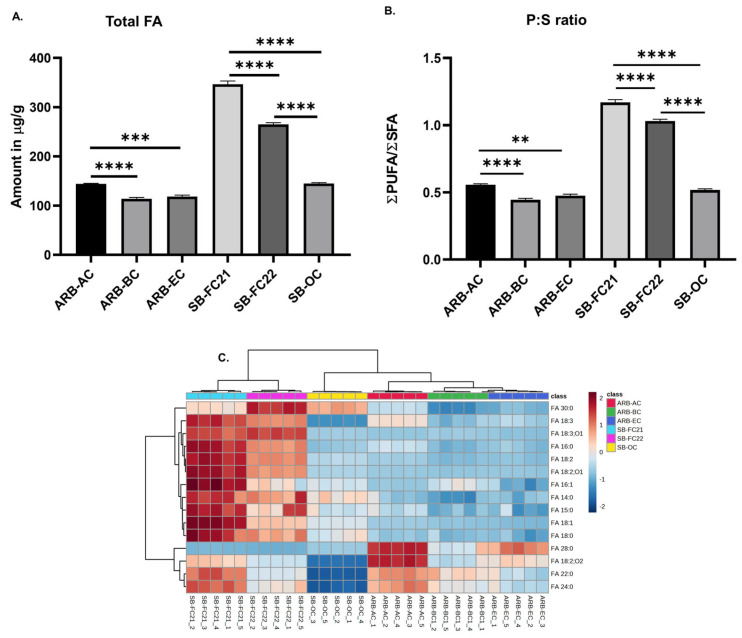
(**A**) Concentrations of free fatty acids (sum of individuals) in six different cultivars of beans. (**B**) PUFA to SFA (P:S ratio). (**C**) Hierarchical cluster correlation analysis of free fatty acids. One-way ANOVA and Tukey’s multiple comparisons test were applied (** *p* < 0.01, *** *p* < 0.001, **** *p* < 0.0001, *n* = 5).

**Figure 3 foods-12-02784-f003:**
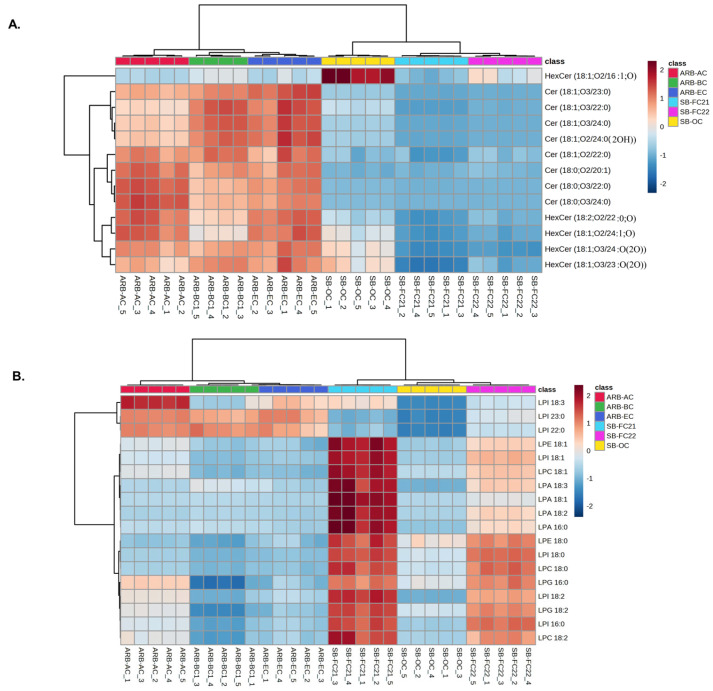
Cluster correlations of (**A**) sphingolipids including ceramides (Cer) and hexosylceramides (HexCer), (**B**) lysophospholipids including LPA, LPC, LPE, LPG (lysophosphatidylglycerol) and LPI (lysophosphatidylinositol) in six different cultivars of beans and their hierarchical cluster analysis.

**Figure 4 foods-12-02784-f004:**
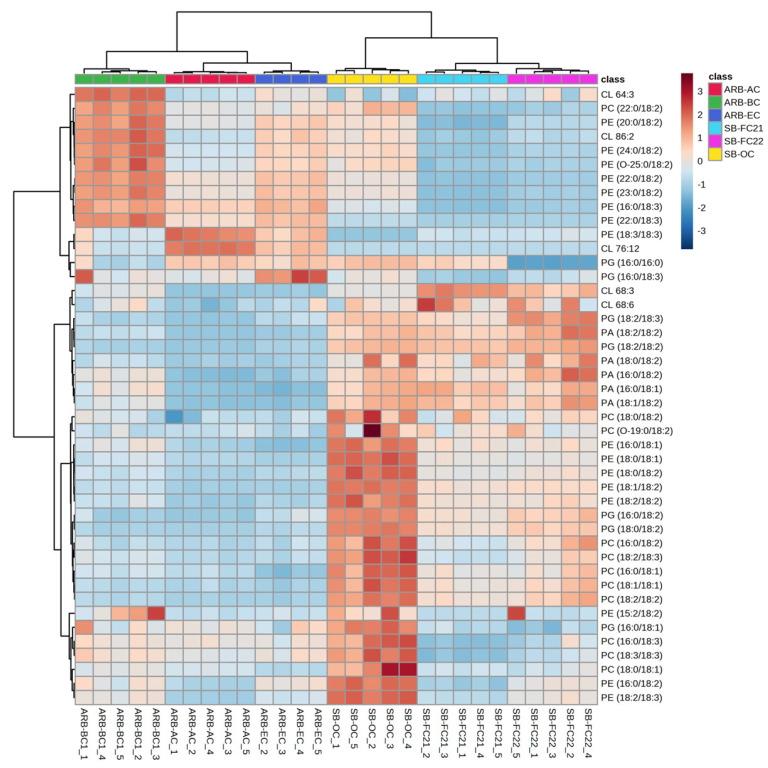
Hierarchical cluster correlation analysis of phosphatidylcholine (PC), phosphatidylethanolamine (PE), phosphatidic acid (PA), phosphatidylglycerol (PG), and cardiolipins in six different cultivars of beans.

**Figure 5 foods-12-02784-f005:**
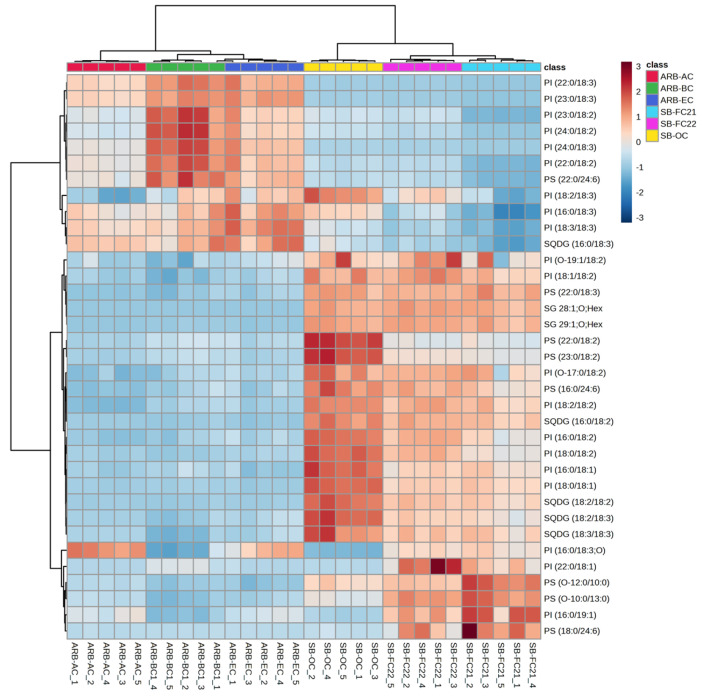
Hierarchical cluster correlation analysis of GPs: phosphatidylinositol (PI) and phosphatidylserine (PS), glycerolipids: sulfoquinovosyl diacylglycerol (SQDG) and sterol glycosides (SGs) in six different cultivars of beans.

**Figure 6 foods-12-02784-f006:**
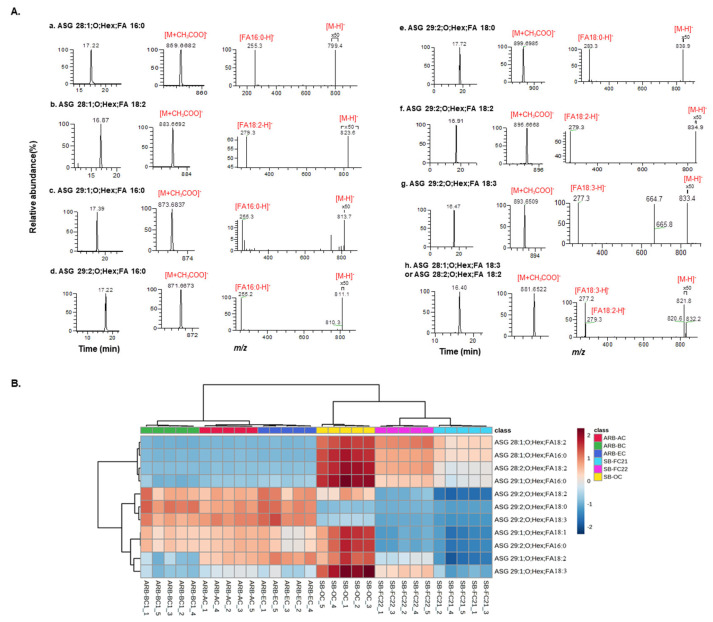
(**A**) Extracted ion chromatogram, MS, and MS/MS spectra of ASGs detected in bean cultivars. (**B**) Hierarchical cluster correlation analysis heatmap of ASG lipid molecular species.

**Table 1 foods-12-02784-t001:** List of identified ASG lipids in adzuki and soybean cultivars.

Sl. No.	Name	RT (min)	*m*/*z*	[M+CH_3_COO]^−^
1	ASG 28:1;O;Hex;FA 16:0	17.22	800.6530	859.6682
2	ASG 28:1;O;Hex;FA 18:2	16.87	824.6530	883.6692
3	ASG 29:1;O;Hex;FA 16:0	17.39	814.6687	873.6837
4	ASG 29:2;O;Hex;FA 16:0	17.22	812.6530	871.6673
5	ASG 29:2;O;Hex;FA 18:0	17.72	840.6843	899.6985
6	ASG 29:2;O;Hex;FA 18:2	16.91	836.6530	895.6668
7	ASG 29:2;O;Hex;FA 18:3	16.47	834.6374	893.6509
8	ASG28:1;O;Hex;FA 18:3 or ASG 28:2;O;Hex;FA 18:2	16.40	823.6407	881.6522

## Data Availability

The datasets generated for this study are available on request to the corresponding author.

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
