# Peer review of "Lipid Composition Analysis and Characterization of Acyl Sterol Glycosides in Adzuki and Soybean Cultivars by Non-Targeted LC-MS"

_foods, 2023, doi:10.3390/foods12142784_

Round 1
Reviewer 1 Report
Please see the attached file and address all my comments.

Moderate editing of English language is required as well as the consistency and spacing check.
Author Response
Response letter attached.

Reviewer 2 Report
The authors developed a Lipid analysis for Adzuki and Soybean via LC-MS. The article is interesting and well written. The novelty is well achieved. However, many revisions are required.
1- In the title correct LC/MS to be LC-MS
2- The new findings and merits for the new method over the old studies in literature should be highlighted in more clear way in the abstract.
3- The following articles should be cited in the introduction
Nutritional Composition, Efficacy, and Processing of Vigna angularis (Adzuki Bean) for the Human Diet: An Overview
https://www.mdpi.com/1420-3049/27/18/6079
Determination of phospholipids in soybean (Glycine max (L.) Merr) cultivars by liquid chromatography–tandem mass spectrometry
https://doi.org/10.1016/j.jfca.2009.12.015
lines 63-67 should be corrected in the light of the aforementioned articles
4- Figures for Adzuki and Soybean should be added for figures in the main text or in the supplementary materials
5- Reference 43 is incomplete “Liu, Y.; Chen, M.; Li, Y.; Feng, X.; Chen, Y.; Lin, L. Liquid Chromatography – Time-of-Flight Tandem Mass Spectrometry. 2022”
6- In line 76, the geographical origins for the three different cultivars should be stated.
7- Lipid extraction 2.3 needs appropriate reference.
8- In line 134, in five different cultivars of beans while in line 145, The lipid composition of all six cultivars ; correct please
9- Full names for SG 28:1; O; Hex, SG 29:1; O; Hex, and PG (18:2/18:2), should be stated.
10- Abbreviation list is strongly recommended
11- Surprisingly, no tables are present. Item 3.5 can be represented in a table form .
12- Figure 6 should be relocated to be in supplementary material to be displayed with the original size chromatograms for more readability.
13- Conclusions are interesting. However, it should be written with full names without abbreviations.
Best wishes
Author Response
Response letter attached

Round 2
Reviewer 1 Report
The new Figure 6 is still incorrect. Please replace [M-CH3COOH]- with [M-H]-.
Note: M generally means molecular mass. Thus, [M-CH3COOH]- means loss of CH3COOH mass from the molecular mass which is not true in the MS/MS results in Figure 6. The corrected one should be loss of CH3COOH from [M+CH3COO]- which is [M-H]-.
Author Response
Comment: The new Figure 6 is still incorrect. Please replace [M-CH3COOH]- with [M-H]-.Note: M generally means molecular mass. Thus, [M-CH3COOH]- means loss of CH3COOH mass from the molecular mass which is not true in the MS/MS results in Figure 6. The corrected one should be loss of CH3COOH from [M+CH3COO]- which is [M-H]-.
Response: Thank you for your kind suggestion again. We have replaced the [M-CH3COOH]- with [M-H]- ion in the Figure 6 as suggested. Kindly see the revised Figure 6.
Reviewer 2 Report
I appreciate their responses. The authors did all required changes.
Author Response
Comment: I appreciate their responses. The authors did all required changes.
Response: Thank you for your appreciations.